# Physicochemical Characterization and Antibacterial Properties of Carbon Dots from Two Mediterranean Olive Solid Waste Cultivars

**DOI:** 10.3390/nano12050885

**Published:** 2022-03-07

**Authors:** Giuseppe Nocito, Emanuele Luigi Sciuto, Domenico Franco, Francesco Nastasi, Luca Pulvirenti, Salvatore Petralia, Corrado Spinella, Giovanna Calabrese, Salvatore Guglielmino, Sabrina Conoci

**Affiliations:** 1Department of Chemical, Biological, Pharmaceutical and Environmental Sciences, University of Messina, Viale Ferdinando Stagno d’Alcontres, 31, 98168 Messina, Italy; gnocito@unime.it (G.N.); emanueleluigi.sciuto@unime.it (E.L.S.); dfranco@unime.it (D.F.); fnastasi@unime.it (F.N.); salvatore.guglielmino@unime.it (S.G.); 2Department of Chemical Science, University of Catania, Viale A. Doria, 6, 95125 Catania, Italy; luca.pulvirenti@phd.unict.it; 3Department of Drug Science and Health, University of Catania, Viale A. Doria, 6, 95125 Catania, Italy; salvatore.petralia@unict.it; 4Istituto per la Microelettronica e Microsistemi, Consiglio Nazionale delle Ricerche (CNR-IMM) Zona Industriale, VIII Strada 5, 95121 Catania, Italy; corrado.spinella@imm.cnr.it; 5Department of Chemistry “Giacomo Ciamician”, University of Bologna, Via Selmi 2, 40126 Bologna, Italy; 6LabSense beyond Nano, URT Department of Physic, CNR Viale Ferdinando Stagno d’Alcontres, 31, 98168 Messina, Italy

**Keywords:** carbon dots, green synthesis, antibacterial properties, *S. aureus*

## Abstract

Carbon nanomaterials have shown great potential in several fields, including biosensing, bioimaging, drug delivery, energy, catalysis, diagnostics, and nanomedicine. Recently, a new class of carbon nanomaterials, carbon dots (CDs), have attracted much attention due to their easy and inexpensive synthesis from a wide range of precursors and fascinating physical, chemical, and biological properties. In this work we have developed CDs derived from olive solid wastes of two Mediterranean regions, Puglia (CDs_P) and Calabria (CDs_C) and evaluated them in terms of their physicochemical properties and antibacterial activity against *Staphylococcus aureus* (*S. aureus*) and *Pseudomonas aeruginosa* (*P. aeruginosa)*. Results show the nanosystems have a quasi-spherical shape of 12–18 nm in size for CDs_P and 15–20 nm in size for CDs_C. UV–Vis characterization indicates a broad absorption band with two main peaks at about 270 nm and 300 nm, respectively, attributed to the π-π* and n-π* transitions of the CDs, respectively. Both samples show photoluminescence (PL) spectra excitation-dependent with a maximum at λ_em_ = 420 nm (λ_exc_ = 300 nm) for CDs_P and a red-shifted at λ_em_ = 445 nm (λ_exc_ = 300 nm) for CDs_C. Band gaps values of ≈ 1.48 eV for CDs_P and ≈ 1.53 eV for CDs_C are in agreement with semiconductor behaviour. ζ potential measures show very negative values for CDs_C compared to CDs_P (three times higher, −38 mV vs. −18 mV at pH = 7). The evaluation of the antibacterial properties highlights that both CDs have higher antibacterial activity towards Gram-positive than to Gram-negative bacteria. In addition, CDs_C exhibit bactericidal behaviour at concentrations of 360, 240, and 120 µg/mL, while lesser activity was found for CDs_P (bacterial cell reduction of only 30% at the highest concentration of 360 µg/mL). This finding was correlated to the higher surface charge of CDs_C compared to CDs_P. Further investigations are in progress to confirm this hypothesis and to gain insight on the antibacterial mechanism of both cultivars.

## 1. Introduction

Since their serendipitous discovery as byproducts of the arc-discharged synthesis of single-walled carbon nanotubes [1], carbon dots (CDs) represent an emerging, luminescent, carbon-based nanomaterial. A simple synthetic procedure, wide precursors, and fascinating physical and chemical properties have successfully stimulated researchers in the last years. In the spreading field of nanotechnology, CDs have been utilized as promising tools for many applications [2], such as optical sensors for ions and molecular species [3], photocatalysis [4], optoelectronics [5], biomaterials [6,7], bio-imaging [8], cancer diagnosis and therapy [9], drug delivery in tumours [10,11], and so on.

CDs are quasi-spherical, carbonaceous nanoparticles with sizes generally below 10 nm. They are mainly constituted by a crystalline sp^2^ core, surrounded by sp^3^ imperfections and high oxygen content on their surface [12]. They are chemically stable, low in toxicity, biocompatible, good conductors/semiconductors, and possess bright luminescence, high photostability, and broadband UV absorption [13]. Their surface is mainly rich with hydroxyls (–OH) and carboxyl/carboxylates (–COOH/–COO^−^). The latter contribute from 5 to 50% (weight) of their oxygen content and impart excellent water colloidal dispersibility and subsequent easy functionalization or passivation with a great variety of chemical species [14]. All these properties can be modulated by synthetic conditions that produce photophysical behaviour, size, and reactivity. CDs present emission bands that are shift-dependent from excitation wavelength modulated by precursor change, surface passivation or heteroatom doping [15]. The reason for this phenomenon is attributed to both (a) the nanometric size that induces the quantum confinement effect and (b) the chemical composition referred to different surface functionalization groups and π-domain extension, inducing many possible states slightly different in energy between the frontier’s orbitals [16].

The synthetic strategies for CD production could be broadly divided into top-down and bottom-up approaches. The first uses physical and chemical methods starting from a wide range of natural or chemical precursors that assemble to produce CDs, such as pyrolysis, hydrothermal treatments, microwave irradiation, and ultrasound. The second uses physical methods to nano-fragment larger, inorganic carbon precursors (graphene, graphite, carbon nanotubes) such as laser ablation, arc discharge, and electrochemistry [9,17,18]. Recently great attention has been also paid to new synthetic methods in terms of green chemistry that address the production of CDs from biomass wastes, cheap or abundant, heterogeneous and biodegradable materials obtained from the manufacturing processes of food, forestry, energy, and many other industrial processes [19]. CDs derived from biomass are greener and, in some cases, better than their chemical counterparts [20] and have been produced using top-down approaches from a lot of precursors, such as papaya [21], spent tea [22], watermelon peels [23], peanut shells [24], wool [25], strawberries [26], olive pits [27], and many others. Further, olive waste management is one of the main ecological issues in the Mediterranean basin, due to the concentration of more than 98% of global olive production and a market in huge expansion over the last two decades [28]. Consequently, their use as precursors for the synthesis of value-added nanomaterials could represent an eco-friendly, economical, and highly available strategy for several applications.

More specifically to the biomedical field, CDs gained a growing interest due to their excellent photoluminescence properties, diverse surface functions, good water solubility, low cytotoxicity, cellular uptake, biocompatibility, microbial adhesion, and theranostic properties [29,30]. Among these properties, antibacterial activity is one of the most appealing features in the design of new biomaterials in which nanotechnology is making fundamental contributions [31,32,33,34]. In this context, the specific physicochemical properties of CDs (e.g., size and surface charge) make them promising tools for addressing antibacterial processes, such as drug resistance, biofilms, and intracellular active/latent bacteria [35,36,37,38,39].

In this context, we have developed CDs derived from the olive solid wastes of two Mediterranean regions, Puglia (CDs_P) and Calabria (CDs_C) and evaluated them in terms of their physicochemical properties and antibacterial activity against *Staphylococcus aureus (S. aureus)* and *Pseudomonas aeruginosa* (*P. aeruginosa*).

## 2. Materials and Methods

### 2.1. Carbon Dots Preparation

CDs were prepared from olive solid wastes collected from two regions of Southern Italy, Puglia (CDs_P) and Calabria (CDs_C), according to the method reported in [40]. Briefly, olive solid wastes were washed several times with boiling water, dried overnight in an oven, and pyrolyzed in a muffle furnace in the absence of air at 600 °C for 1 h. The resulting carbon-based material was finely ground in a mortar and suspended in deionized water (10 mg/mL). The mixture was sonicated in an ultrasonic bath (Bandelin Sonorex RK 100 H, Bandelin electronic GmbH & Co. KG, Berlin, Deutschland) for 10 min; 1 mL of hydrogen peroxide (H_2_O_2_ sol. 30% (*w*/*w*)—Sigma-Aldrich, Milan, Italy) was added, and then it was refluxed for 90 min under stirring. The reaction mixture was purified by centrifugation at 8000 rpm for 20 min (Eppendorf Centrifuge 5430, Eppendorf SE, Hamburg, Deutschland), and the supernatant was syringe-filtered (Sartorius Minisart RC 0.2 μm, Sartorius AG, Göttingen, Deutschland). Lastly, the concentration of CDs’ colloidal dispersion was estimated by weighing (Sartorius Quintix balance, Sartorius AG, Göttingen, Deutschland) after evaporation of the solvent under reduced pressure and subsequent drying in the oven (CDs_P ≈ 0.7 mg/mL; CDs_C ≈ 1.4 mg/mL). The mixture was simply purified by centrifugation and filtration, obtaining the final CD colloidal dispersion. A production yield of about 10% was obtained from each olive solid waste cultivar.

### 2.2. Chemical and Physical Characterization

UV–Vis absorption spectra were recorded with a Jasco V-560 spectrophotometer, (JASCO Corporation, Tokyo, Japan), steady-state photoluminescence spectra with a Spex Fluorolog-2 (mod. F-111) spectrofluorometer (Horiba Ltd., Kyoto, Japan) in air-equilibrated 1 cm quartz cells.

The isoelectric points of the CDs’ colloidal dispersions were estimated by ζ (Zeta) potential pH titration using the dynamic light scattering (DLS) technique with a Malvern Zetasizer Nano ZS90 instrument (Malvern Panalytical Ltd., Malvern, United Kingdom). pH was moved to 8, 10, and 2 by adding, respectively, NaOH 0.1 M (sodium hydroxide—Sigma-Aldrich, Merck KGaA, Darmstadt, Deutschland) and HCl 0.1 M solutions (hydrochloric acid—Sigma-Aldrich, Milan, Italy). The isoelectric point was found by plotting the pH vs. Zeta potential and intercepting the pH value when the ζ potential was zero.

X-ray photoelectron spectroscopy (XPS) was performed on carbon dots deposited on silicon slides using a PHI 5600 multi-technique ESCA-Auger spectrometer (Physical electronics Inc., Chanhassen, MN, USA) equipped with a monochromatic Al-Kα X-ray source. The XPS binding energy (BE) scale was calibrated on the C 1s peak of adventitious carbon at 285.0 eV. Transmission FT–IR measurements on the silicon-deposited carbon dots were obtained using a JASCO FTIR 4600LE spectrometer (JASCO Corporation, Tokyo, Japan) in the spectral range of 560–4000 cm^−1^(resolution 4 cm^−1^).

Transmission electron microscopy (TEM) analysis was performed using the bright field in conventional TEM parallel beam mode. An ATEM JEOL JEM 2010 equipped with a 30 mm^2^ window energy dispersive X-ray (EDX) spectrometer (JEOL Ltd., Musashino, Akishima, Tokyo, Japan), was used.

### 2.3. Bacterial Assays

*S. aureus* (ATCC 29213) was purchased from American Type Culture Collection (LGC Promochem, Milan, Italy) and cultured in tryptone soya broth (TSB, Sigma-Aldrich, Milan, Italy). *P. aeruginosa* (ATCC 27853) was purchased from American Type Culture Collection (LGC Promochem, Milan, Italy) and cultured in Luria–Bertani broth (LB, Sigma-Aldrich, Milan, Italy).

Antibacterial tests were performed in Mueller–Hinton broth (MHB, Sigma-Aldrich, Milan, Italy), a culture medium susceptible to antibiotics.

To evaluate the minimal inhibitory concentration (MIC) of both CDs’ colloidal dispersions, the microplate inhibition assay was used. Specifically, semi-exponential cultures of bacterial strains at the final concentration of about 10^5^ bacteria per mL were inoculated in MHB in the presence of increasing concentrations of CDs (60–360 µg/mL) in 96-well plates and incubated at 37 °C under shaking overnight. After incubation, the concentrations inhibiting at least 90% and 99.9% of bacteria, MIC_90_ and MIC_99,_ were determined compared to the untreated control.

To evaluate the bacterial cell viability, an MTS assay (CellTiter 96^®^ AQueous One Solution Cell Proliferation Assay, Promega, Milan, Italy) was performed. In more detail, bacterial cultures in the presence of different concentrations of CDs were grown overnight at 37 °C in 96-well plates. Then, MTS reagent was added to the bacteria culture media, incubated for 2 h at 37 °C in static condition, the plate was shaken briefly, and absorbance was measured at 490 nm by using a microtiter plate reader (Multiskan GO, Thermo Scientific, Waltham, MA, USA). The reduction of bacterial viability was evaluated in terms of the percentage of MTS reduction (% MTS_red_), compared to the untreated control (CTR) using the following equation:(1)$${\mathrm{MTS}}_{\mathrm{red}}\left(\%\right)=\left(\frac{A}{B}\right)\times 100$$
where *A* e *B* are the OD_490_ from the MTS-reduced formazan of condition with CDs and CTR. The samples were analysed in triplicate for each experimental condition.

Figure 1 reports the schematic representation of CD preparation, physico-chemical characterization, and bacterial testing.

## 3. Results and Discussion

### 3.1. Physicochemical Characterization of CDs

The optical properties of both synthesized CDs were characterized by the UV–Vis absorbance and photoluminescence emission spectra displayed in Figure 2. The absorption spectrum of CDs_P (Figure 2a) exhibits broadband UV absorption, with a trend compatible with the light scattering operated by a small nanoparticle’s colloidal dispersion. The UV–Vis absorbance spectrum of CDs_P shows two detectable peaks as shoulders located around 270 nm and 300 nm, attributed respectively to the π–π* transition of CDs and n–π* transitions of the functional groups present on CDs [41,42]. These two absorption shoulders probably suggest the existence of conjugated structures as well as the presence of functional groups containing oxygen in the CDs [12,13,14]. Figure 2b shows the PL spectra of CDs_P with the excitation wavelength in the interval between 300 nm and 450 nm. The emission maximum is around λ_em_ = 420 nm for λ_exc_ = 300 nm.

Although the absorption spectrum of CDs_C showed a trend similar to that exhibited by CDs_P, some differences are present (Figure 2a–c). First, the peaks detected as shoulders are shifted to lower energies, founding them around 280 nm and 320 nm (Figure 2c). Secondly, although CDs_C show PL, the emission spectra registered in the same excitation wavelength interval (300–450 nm) are red-shifted with respect to CDs_P, as for the absorption bands [43]. The emission maximum is around λ_em_ = 445 nm for λ_exc_ = 300 nm (Figure 2d).

To further evaluate the difference between CDs derived from two different Mediterranean olive solid waste cultivars, we calculated the respective band gaps from UV–Vis absorption spectra reported in Figure 2a,c, using a Tauc plot [44] with the formula [45]:(2)$${\left(\alpha h\nu \right)}^{\frac{1}{\gamma }}=B\left(h\nu -E_{g}\right)$$
where α is the absorption coefficient ($\alpha =2.303\;{\mathrm{Acm}}^{-1}$), h is the Plank constant, ν is the frequency of the incident photon. The γ factor depends on the nature of the electronic transition, and in our case for permitted transition, it could be 0.5 for the indirect one and 2 for the direct one. B is a constant assumed to be 1, and $E_{g}$ is the energy band gap. $E_{g}$ was calculated using $\gamma =\frac{1}{2}$ for direct electronic transition.

Plots are reported in Figure 3. Band gap values of ≈ 1.48 eV for CDs_P and ≈ 1.53 eV for CDs_C, respectively, were found. These values are in agreement with semiconductor behaviour, according to similar values found in [46,47].

Concerning the chemical characterization of CDs_P, both FTIR and XPS analysis were reported in our previous works [40,48] and showed for FTIR the following peaks: 3424, 3236, 2923/2850, 1656, 1412, 1320, 1116, and 1096 cm^−1^, attributed to –OH, N–H, C–H, C=O (carbonyl), COO^−^ (carboxylate), C–OH (hydroxyl), and C–O–C (epoxide) groups, respectively. XPS analysis showed peaks of C1s at 285 eV for C–C and 289 eV for O=C–O respectively. Regarding CD_C, FTIR analysis (see Appendix A) reveals the presence of –OH, C=O (carbonyl), COO^−^ (carboxylate), C–OH (hydroxyl), and C–O–C (epoxide) groups, while the XPS spectrum (see Appendix A) exhibits the same C1s peaks of CDs_P at a 285 eV for C–C and 289 eV for O=C–O, respectively. The similarity of surface groups for both cultivars also account for similar band gap values (see above).

Figure 4 reports the plot of ζ potential values as a function of pH for both CDs_P and CDs_C. It can be noticed that both cultivars show negative surface charges, but CDs_P exhibits higher values of ζ potential with respect to CDs_P. According to that, the values of the isoelectric point (pH value at 0 charge) correspond to pH ≈ 3 for CDs_P and pH ≈ 2.4 for CDs_C. More interesting, in the physiological conditions (pH ≈ 7), CDs_C feature a charge (ζ potential) about three times more negative than CDs_P (−32 mV vs. −11 mV). This certainly can be a notable point for the antibacterial activity of the nanomaterials (vide infra).

Morphological characteristics were also investigated by transmission electron microscopy (TEM). Figure 5 reports TEM images of the two different cultivars. CDs_P displays dispersed quasi-spherical nanoparticles with particle size ranging between 12–18 nm, while CDs_C exhibits similar characteristics, with a mean size of 15–20 nm.

### 3.2. Antibacterial properties of Carbon Dots

The bactericidal activity of both CD dispersions was evaluated against both *S. aureus* (Gram-positive) and *P. aeruginosa* (Gram-negative), two of the most common pathogens involved in a wide range of infections. First, values of MIC for both CD types were determined by microplate inhibition assay up to the maximum concentration of 360 µg/mL. In Table 1 are reported the MIC_90_ and MIC_99_ values [µg/mL] for both CDs.

Our results showed that CDs_C exhibit an *S. aureus* viability reduction of about 99.9% (MIC_99_) at 360 µg/mL and 90% bacterial inhibition (MIC_90_) at about 120 µg/mL, while no bacterial inhibition was observed against *P. aeruginosa.* On the contrary, no bacterial inhibition was observed for CDs_P at all using concentrations (ranging from 60 to 360 µg/mL) against either bacterial strain.

To further evaluate the antibacterial properties of both CD types, we also performed an MTS cell viability assay (Figure 6 and Figure 7).

MTS data showed that CDs_C dispersion reduces almost completely *S. aureus* viability at the higher concentrations (360, 240, and 120 µg/mL), while at the lowest concentration (60 µg/mL), it is reduced by about 50% compared to the control (untreated). On the contrary, MTS data of CDs_P indicated a bacterial cell reduction of only 30% at the highest concentration (360 µg/mL), showing very poor antibacterial activity. In more details, the bacterial viability of CDs_C was found of 47.5 ± 2.37% at 60 µg/mL, 4.75 ± 0.24% at 120 µg/mL, 0.47 ± 0.005% at 240 µg/mL, and 0.05 ± 0.0005% at 360 µg/mL; for CDs_P, it was 94.9 ± 4.74% at 60 µg/mL, 92.6 ± 4.63% at 120 µg/mL, 85.7 ± 4.28% at 240 µg/mL, and 68.4 ± 3.42% at 360 µg/mL. These data were in agreement with MIC results.

On the other hand, MTS data obtained against *P. aeruginosa* showed that only CDs_C at the higher concentration (360 µg/mL) exhibited a bacterial reduction of about 20%. In more detail, the bacterial viability of CDs_C was found to be 98.4 ± 1.17% at 60 µg/mL, 95.75 ± 1.49% at 120 µg/mL, 93.43 ± 1.12% at 240 µg/mL, and 83.81 ± 3.5% at 360 µg/mL.

In addition to the above considerations, we also observed a different level of antibacterial activity between CDs_P and CDs_C, more evident in *S. aureus* than in *P. aeruginosa* probably due to the different cell wall compositions. Several action mechanisms have been suggested to explain the antibacterial activity of CDs closely related to their physicochemical properties, including their dimensionalities, lateral size, shape, number of layers, surface charges, the presence and nature of surface functional groups, and doping [35,36,37,38].

A hypothesis to explain the dissimilar antibacterial behaviour of the two CD types could be the charge distribution and functional groups of the CDs. Recently, some studies suggested that factors due to particle size and surface functionalization and charges can affect antibacterial effects [35,49]. With specific focus on surface charge, Bing et al. studied the antibacterial capability of CDs with three different surface charges (uncharged, positive, and negative) finding only positive- (spermine derived positive CDs, SC-dots) and negative-charged CDs (candle-soot derived negative CDs, CC-dots) exhibited signs of cell death on *E. coli* (Gram negative), such as DNA fragmentation, extracellular exposure of phosphatidylserine, condensation of the chromosome, and loss of structural integrity, but the same effects did not occur with uncharged glucose carbon dots (GC-dots). Based on these considerations, our results could be compared with negatively charged candle-soot C-dots (CC-dots). Although similar results were obtained against *P. aeruginosa,* we found that *S**. aureus* viability was almost totally and slightly reduced when treated with CDs_C and CDs_P, respectively. Since our systems, CDs_C and CDs_P, exhibit similar sizes of nanoparticles, and no surface functionalization was carried out, neither factor is relevant to the findings, indicating that CDs_C show higher antibacterial activity compared with to CDs_P. These results suggest, rather, different antibacterial mechanisms for the two cultivars, probably attributable to the different surface charges and ζ potential. Although both CDs from the two cultivars exhibited negative surface charges, CDs_C have much more of a negative charge than do CDs_P (about −38 mV for CDs_C and −18 mV for CDs_P at pH = 7). Actually, during the bacterial growth, the medium pH decreased over time (0–8 h), reaching more acidic values (from 7.2 to about 5.5) (See Appendix A). By considering this aspect, we can observe that the surface charge of both cultivars reaches about −32 mV (CDs_C) and −11 mV (CDs_P), respectively, at pH = 5.5, highlighting that CDs_C feature a negative surface charge 3 times higher than that of CDs_P. This is probably the reason for the increased antibacterial activity of CDs_C in contrast with CDs_P. Further studies are in progress to validate our hypothesis and gain insights on the active antibacterial mechanism.

## 4. Conclusions

In this paper, we report on the development of CDs derived from olive solid wastes of two Mediterranean regions, Puglia (CDs_P) and Calabria (CDs_C) and their evaluation in terms of their physicochemical properties and antibacterial activity against *S. aureus*. The UV–Vis characterization shows a typical broad absorption band with two main peaks at about 270 nm and 300 nm, attributed to the π–π* and n–-π* transitions of CDs, respectively. The PL spectra are excitation-dependent, and CDs_P shows emission maximum at λ_em_ = 420 nm (λ_exc_ = 300 nm), while CDs_C has an emission maximum red-shifted at λ_em_ = 445 nm (λ_exc_ = 300 nm). Band gaps values of ≈ 1.48 eV for CDs_P and ≈ 1.53 eV for CDs_C, respectively, were found, in agreement with semi-conductor behaviour. ζ potential values are negative for both cultivars. However, CDs_C feature a more negative charge, in fact three times more negative than that of CDs_P (−38 mV vs. −18 mV at pH = 7). The TEM morphological inspection shows quasi-spherical nanoparticles 12–18 nm in size for CDs_P and 15–20 nm in size for CDs_C. The evaluation of antibacterial properties highlights that both CDs have higher antibacterial activity towards Gram-positive bacteria than to Gram-negative bacteria. In addition, the evaluation of the antibacterial properties towards *S. aureus* of both CD types highlights that CDs_C exhibit antibacterial properties at concentrations of 360, 240, and 120 µg/mL, while CDs_P shows a bacterial cell reduction of 30% at the highest concentration of 360 µg/mL. This finding was correlated to the highest surface charge of CDs_C compared to CDs_P, which is still very negative during bacterial growth, reaching an acidic pH of 5.5. Further investigations are in progress to confirm this hypothesis and gain insights on the antibacterial mechanism of both cultivars.

## Figures and Tables

**Figure 1 nanomaterials-12-00885-f001:**
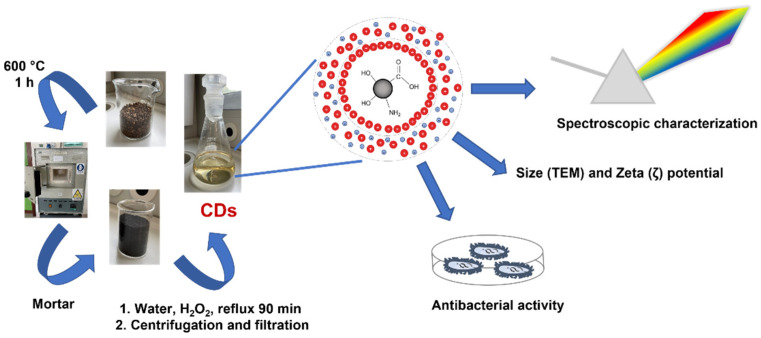
Schematic representation of CD preparation, physico-chemical characterization, and bacterial testing.

**Figure 2 nanomaterials-12-00885-f002:**
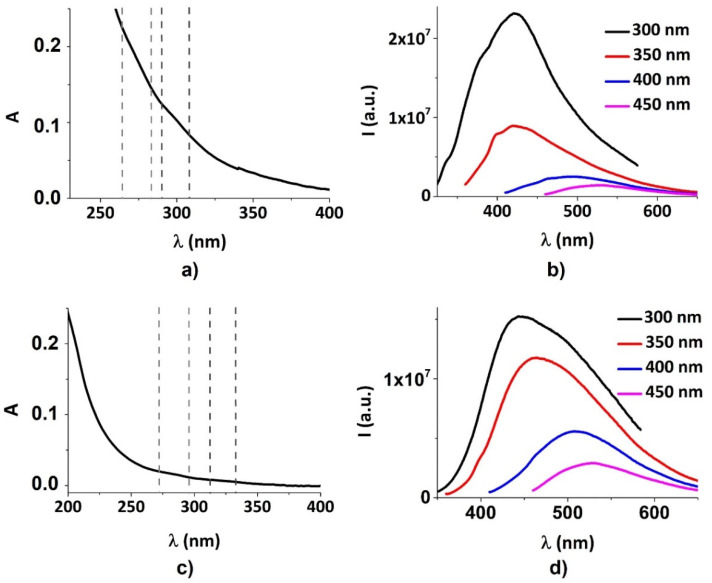
Optical characterization of CDs: (**a**) UV–Vis absorption spectrum of CDs_P; (**b**) Photoluminescence spectra of CDs_P; (**c**) UV–Vis absorption spectrum of CDs_C; (**d**) Photoluminescence spectra of CDs_C.

**Figure 3 nanomaterials-12-00885-f003:**
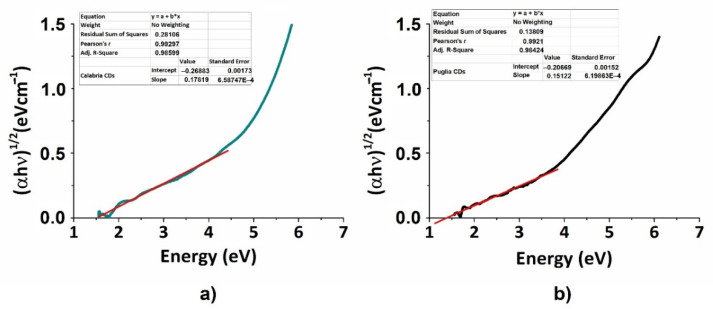
Tauc plots for the optical energy band gap calculation: (**a**) CDs_P; (**b**) CDs_C.

**Figure 4 nanomaterials-12-00885-f004:**
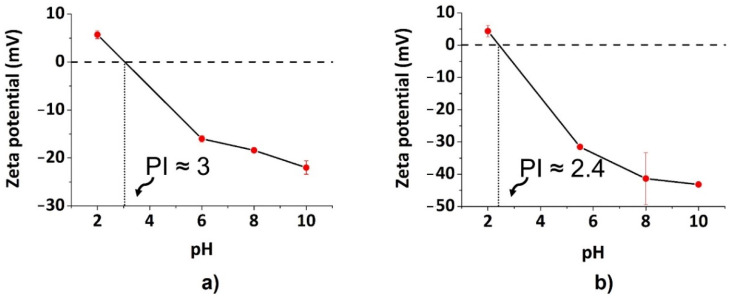
Graphical representation of ζ potential as a function of pH titration: (**a**) CDs_P; (**b**) CDs_C.

**Figure 5 nanomaterials-12-00885-f005:**
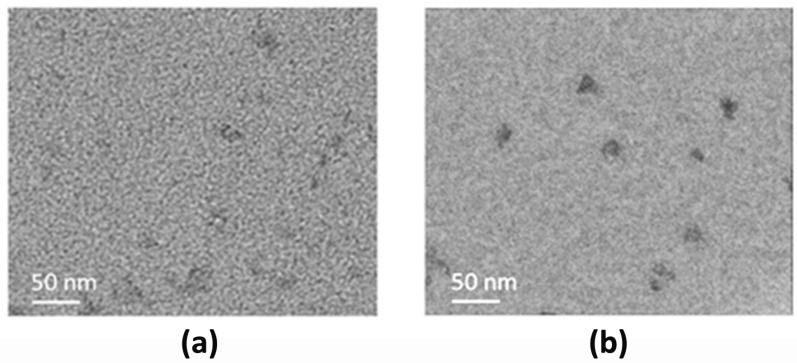
Representative transmission electron microscopy (TEM) images of: (**a**) CDs_P; (**b**) CDs_C.

**Figure 6 nanomaterials-12-00885-f006:**
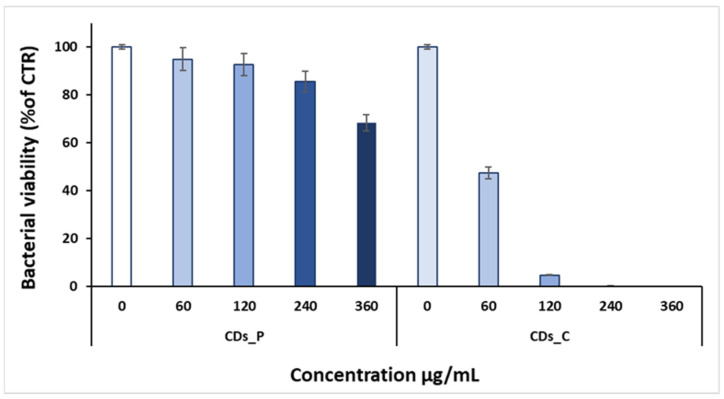
MTS assay of CDs_C and CDs_P against *S. aureus* strain. Data are presented as the mean ± SD from three independent experiments.

**Figure 7 nanomaterials-12-00885-f007:**
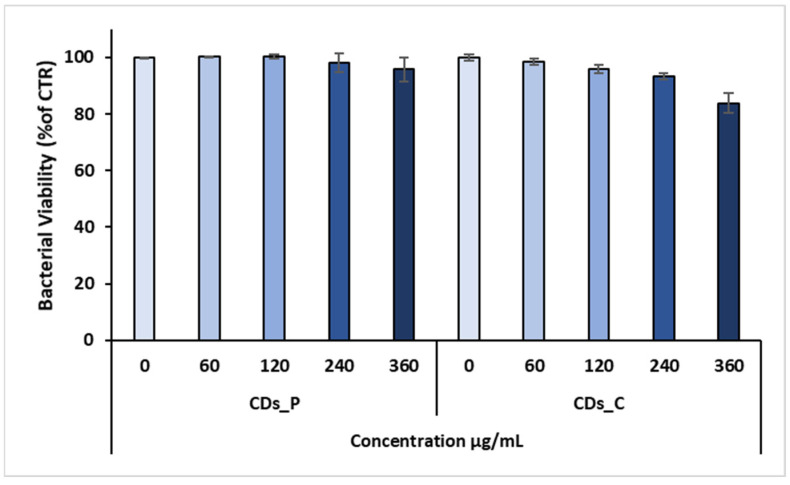
MTS assay of CDs_C and CDs_P against *Pseudomonas aeruginosa* strain. Data are presented as the mean ± SD from three independent experiments.

**Table 1 nanomaterials-12-00885-t001:** MIC values [µg/mL] of *S. aureus* strains in Mueller–Hinton broth (MH).

Bacterial strain	CDs	MIC_90_	MIC_99_
*S. aureus*	CDs_P	-	-
	CDs_C	120 µg/mL	360 µg/mL
*P. aeruginosa*	CDs_P	-	-
	CDs_C	-	-
Dash (-) = no antibacterial activity up to the concentration of 360 µg/mL.			

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
