# Peer review of "Physicochemical Characterization and Antibacterial Properties of Carbon Dots from Two Mediterranean Olive Solid Waste Cultivars"

_nanomaterials, 2022, doi:10.3390/nano12050885_

Round 1
Reviewer 1 Report
The paper focuses on the synthesis of Carbon dots derived from olive solid waste. The nanoparticles have been characterised via TEM microscopy and zeta potential pH titration, their optical properties have been studied via UV-vis and fluorescence spectroscopy and their antimicrobial behavior against S. aureus has been assessed . The paper falls within the scope of the journal. My comments/suggestions are shown below:
- The nanoparticle characterization is incomplete: FTIR , XPS and elemental analysis are essential. Note that the discussion about the antimicrobial mechanism of Carbon dots highlights the role of surface functionalities, but no FTIR is provided.
- The last sentence in Introduction “Results are presented and discussed” is redundant
- In “Carbon dots preparation” the author stated “The resulting carbon was finely grounded”- Do the author indicate that their materials are composed of 100% carbon?
- What is the yield of the preparation method?
- The synthesis relies on pyrolysis at 600 for 1 h in the absence of air. Why was such a high pyrolysis temperature (and thus a highly energy intensive procedure) was used? Did the authors explore lower pyrolysis temperature?
- The antimicrobial behavior of Carbon dots has been tested only against S. aureus (a gram positive type of bacteria), however it should also be tested against a representative class of gram-negative bacterial.
- The antimicrobial behavior of Carbon dots reported here should be compared with related systems reported previously.
Author Response
Response to Reviewer 1 Comments
The paper focuses on the synthesis of Carbon dots derived from olive solid waste. The nanoparticles have been characterised via TEM microscopy and zeta potential pH titration, their optical properties have been studied via UV-vis and fluorescence spectroscopy and their antimicrobial behavior against S. aureus has been assessed. The paper falls within the scope of the journal. My comments/suggestions are shown below:
Reviewer: The nanoparticle characterization is incomplete: FTIR, XPS and elemental analysis are essential. Note that the discussion about the antimicrobial mechanism of Carbon dots highlights the role of surface functionalities, but no FTIR is provided.
Response: We thank the reviewer for the comment. About CDs_P, FTIR analysis was reported in our previous works [40,46] and showed the following peaks at 3424, 3236, 2923/2850, 1656, 1412, 1320 and 1116 and 1096 cm−1, attributed to –OH, N-H, C-H, C=O (carbonyl), COO- (carboxylate), C-OH (hydroxyl) and C-O-C (epoxide) groups, respectively. About CD_C, FTIR analysis we included it in supplementary materials [S1] and the spectrum reveals the presence of –OH, C=O (carbonyl), COO- (carboxylate), C-OH (hydroxyl) and C-O-C (epoxide) groups.
We include in the manuscript a discussion about FTIR.
Reviewer: The last sentence in Introduction “Results are presented and discussed” is redundant
Response: Sentence has been deleted.
Reviewer: In “Carbon dots preparation” the author stated “The resulting carbon was finely grounded”- Do the author indicate that their materials are composed of 100% carbon?
Response: We thank the reviewer for the suggestion. To better explain the composition of the starting material, the sentence has been modified as “The resulting carbon-based material was finely grounded in a mortar and suspended in deionized water (10 mg/ml)”.
Reviewer: What is the yield of the preparation method?
Response: The yield of the preparation method is approximately 10%. Additional information has been included in the revised version of manuscript and highlighted in yellow.
Reviewer: The synthesis relies on pyrolysis at 600 for 1 h in the absence of air. Why was such a high pyrolysis temperature (and thus a highly energy intensive procedure) was used? Did the authors explore lower pyrolysis temperature?
Response: The pyrolysis for 1 h at 600 °C was chosen based on our previously optimized protocols [40]. We haven’t tested lower pyrolysis temperatures, but we thank the reviewer for the suggestion which we will consider in future activities.
Reviewer: The antimicrobial behaviour of Carbon dots has been tested only against S. aureus (a gram positive type of bacteria), however it should also be tested against a representative class of gram-negative bacterial.
Response: As suggested by the reviewer, we also evaluated CDs against P. aeruginosa, as a representative class of Gram-negative bacteria. Data, a new figure (Figure 7) and discussion have been included in the revised manuscript.
Reviewer: The antimicrobial behaviour of Carbon dots reported here should be compared with related systems reported previously.
Response: A comparative discussion with related systems reported previously has been added in the revised version of manuscript.
Reviewer 2 Report
1) Is there any similar for your antibacterial activity compared with those in Ref.[45] ? ....page 7, L 233.
2) What is GC-dots [46]? ....page 7, L 238
3) Spelling check...."such us fractured DNA"?? ....page 7, L236
4) What is the main reason for two kinds of the CD-dots having the rather same optical properties, but show different antibacterial activity ?
Author Response
Response to Reviewer 2 Comments
Reviewer: Is there any similar for your antibacterial activity compared with those in Ref.[45] ? ....page 7, L 233.
Response: About ref. [45, Sun et al.], authors obtained bactericidal activities by using biguanide antimicrobial agent to prepare CDs, while no bactericidal activities were showed by CDs prepared by using citric acid and urea, folic acid, glucose, and orange juice as precursors. On the contrary, in our case, CDs were obtained from olive solid waste precursor and they showed antibacterial activity, especially against S. aureus.
Reviewer: What is GC-dots [46]? ....page 7, L 238
Response: GC-dots is the acronym for uncharged glucose Carbon dots, a CDs obtained by Bing et al., with neutral surface charge. The explanation of the acronym has been included in the revised version of the manuscript.
Reviewer: Spelling check...."such us fractured DNA"?? ....page 7, L236
Response: We check the sentence and change it in “such as DNA fragmentation”. We apologize for the typo.
Reviewer: What is the main reason for two kinds of the CD-dots having the rather same optical properties, but show different antibacterial activity?
Response: Results suggest that the different antibacterial mechanism for the two cultivars is probably attributable to the different surface charges and ζ potential. See discussion in the lines 301-328.
Reviewer 3 Report
The manuscript presents several points to be addressed:
- Previous work, in the same area of research, has been done and must be cited such as: ACS Sustainable Chemistry & Engineering 7 (12), 10493-10500.
- In Line 161 I think is a mistake, must be Figure 1a-c.
- The acronysm CDs must be kept throw the text, such us captions Fig. 1.
- Authors are sure that for CDs_P is 1.84 eV?????, I can watch below 1.5 eV, please explain better this
- 5. In this point, the XPS spectra must be showed, becasue the meaning of the band gap depend of the size and functional groups on the surface of the CDs, and here is missing all this discussion. Authors must explain this important point, please read the suggested paper above about this
The research is work must be improved before publication, attending these points
Author Response
Response to Reviewer 3 Comments
The manuscript presents several points to be addressed:
Reviewer: Previous work, in the same area of research, has been done and must be cited such as: ACS Sustainable Chemistry & Engineering 7 (12), 10493-10500.
Response: The reference has been added in Introduction section and subsequent references renumbered.
Reviewer: In Line 161 I think is a mistake, must be Figure 1a-c.
Response: Thanks to the Reviewer for the observation. We corrected the mistake.
Reviewer: the acronysm CDs must be kept throw the text, such us captions Fig. 1.
Response: We have change CD into CDs in the captions of Figure 1 and checked for other typos in the text.
Reviewer: Authors are sure that for CDs_P is 1.84 eV?????, I can watch below 1.5 eV, please explain better this
Response: There was a typo in the value of CDs_P band gap. The correct value is 1.48 eV instead of 1.84 (we inverted the last two numbers).
Reviewer: In this point, the XPS spectra must be showed, becasue the meaning of the band gap depend of the size and functional groups on the surface of the CDs, and here is missing all this discussion. Authors must explain this important point, please read the suggested paper above about this
Response: We thank the reviewer for the comment. About CDs_P, XPS analysis was reported in our previous works [40,46] and showed the following peaks of C1s at 285eV for C-C and 289 eV for O=C-O respectively. About CD_C, XPS analysis we included it in supplementary materials [S2] exhibits the same C1s at a285eV for C-C and 289 eV for O=C-O respectively. We include in the manuscript a discussion about XPS and band gap (see lines 225-234).
Reviewer 4 Report
Very nice piece of work. I would recommend with minor modifications before acceptance...
- Authors should mention the importance of the precursor used as compared to various green precursors mentioned in the introduction section.
- Section 2 should be shown schematically.
- Results of Figure 2 i.e. band gap should be discussed with results in literature. At least discuss the results of the some literature ...
Author Response
Response to Reviewer 4 Comments
Very nice piece of work. I would recommend with minor modifications before acceptance...
Reviewer: Authors should mention the importance of the precursor used as compared to various green precursors mentioned in the introduction section.
Response: According to Reviewer's suggestion, we better explain the importance of the precursor used in this work in the introduction section.
Reviewer: Section 2 should be shown schematically.
Response: According to Reviewer's suggestion, we added, at the end of section 2, a new figure illustrates a schematic representation of CDs preparation, physico-chemical characterization and bacterial testing.
Reviewer: Results of Figure 2 i.e. band gap should be discussed with results in literature. At least discuss the results of the some literature ...
Response: Thank you for the suggestion, we have added other two literature works in which energy band gap values similar to ours were found.
Round 2
Reviewer 1 Report
The revised manuscript has been improved and it can be published.
Reviewer 3 Report
The research work has been improved accordintg the suggested points